Effects of bamboo biochar on soil physicochemical properties and microbial diversity in tea gardens

Zhang Si-Hai 1
Shen Yi 1
Lin Le-Feng 1
Tang Su-Lei 2
Liu Chun-Xiao 1
Fang Xiang-Hua 3
Guo Zhi-Ping 2 gzplsu@126.com
Wang Ying-Ying 1 wangyingying1015@163.com
Zhu Yang-Chun 2
1 College of Liangshan, Lishui University , Lishui, Zhejiang Province , China
2 College of Ecology, Lishui University , Lishui, Zhejiang Province , China
3 Forestry Science and Technology College, Lishui Vocational and Technical College , Lishui, Zhejiang Province , China
Kent Clement
Electronic publication date: 2024 Dec 5
Publication date: 2024
Volume: 12
Electronic Location ID: e18642
Received 2024 Jul 18; Accepted 2024 Nov 13
Copyright: © 2024 Zhang et al.
Copyright year: 2024
Copyright holder: Zhang et al.
License: This is an open access article distributed under the terms of the Creative Commons Attribution License, which permits unrestricted use, distribution, reproduction and adaptation in any medium and for any purpose provided that it is properly attributed. For attribution, the original author(s), title, publication source (PeerJ) and either DOI or URL of the article must be cited.
License URL: https://creativecommons.org/licenses/by/4.0/

Keywords: Acidic soil, Physicochemical properties, Enzymatic activity, Microbial community, Bamboo biochar

Funding: Public Welfare Technology Application Research 2021GYX11; 2022ZDYF02 Natural Science Foundation of Zhejiang Province LSSY24C150022 This research was funded by the Public Welfare Technology Application Research Project of Lishui city (2021GYX11; 2022ZDYF02), the Natural Science Foundation of Zhejiang Province (LSSY24C150022). The funders had no role in study design, data collection and analysis, decision to publish, or preparation of the manuscript.

==============================
Biochar, a carbon-rich material that has attracted considerable interest in interdisciplinary research, is produced through a process known as pyrolysis, which involves the thermal decomposition of organic material in the absence of oxygen. Bamboo biochar is a specific type of biochar, manufactured from bamboo straw through carbonisation at 800 °C and subsequent filtration through a 100-mesh sieve. There is currently a lack of research into the potential benefits of bamboo biochar in improving soil quality in tea gardens. The aim of this study was to investigate the effect of bamboo biochar on the physicochemical properties, enzymatic activity, and microbial community structure of tea garden soils. The results demonstrate that the integration of bamboo biochar into the soil significantly enhanced the soil pH, total nitrogen, available nitrogen, total phosphorus, available phosphorus, available potassium, and slowly available potassium by 15.3%, 52.0%, 91.5%, 91%, 48.4%, 94.2%, and 107.7%, respectively. In addition, soil acid phosphatase activity decreased significantly by 52.5%. In contrast, the activities of sucrase, catalase, and β-glucosidase increased substantially by 54.0%, 68.7%, and 68.4%, respectively, when organic fertilizer and bamboo biochar were applied concurrently. Additionally, the Shannon, Simpson, and Pielou diversity indices of the microbial communities were significantly enhanced. Following the incorporation of bamboo biochar in the soil samples, the relative abundance of Proteobacteria increased significantly, whereas that of Acidobacteria decreased. Various concentrations of bamboo biochar markedly influenced microbial markers in the soil. The results of this study suggest that the application of bamboo biochar to soil may modestly improve its physicochemical properties, enzyme activity, and microbial community structure. These findings provide a foundation for future investigations on soil ecological restoration.

Introduction

Soil acidification is a significant challenge in soil degradation. The primary causes of acidification in agricultural soils are the inappropriate fertiliser application and cultivation practices (Barak et al., 1997). The process of soil acidification has markedly accelerated in China, particularly in recent decades, owing to the prolonged and excessive use of chemical fertilisers in conjunction with high-intensity human activities (Guo et al., 2010).

Chinese tea is an acidophilic crop, and the optimal soil pH range for its cultivation is 4.5–6.0. A soil pH < 4.5 impedes tea growth, resulting in a reduction in output and quality (Li et al., 2016; Qiao et al., 2018; Yan et al., 2020). The acidification of tea garden soils is a more severe problem than that of other crops because of the long-term and extensive use of chemical fertilisers and tea metabolism (Wang, Li & Xu, 2009). The cultivation of tea plants results in the exudation of organic acids from their roots into the surrounding soil, leading to a gradual decrease in soil pH. This process occurs at a faster rate than in natural soil (Ruan et al., 2023). Consequently, soil acidification has the potential to impair soil fertility, with a notable decline in available nutrients (Song et al., 2022).

Organic fertilisers, sourced from natural materials such as livestock and poultry excreta, plant residues, biogas residues, and agricultural by-products, have been shown to positively influence pollution mitigation. Numerous studies have reported the potential benefits of organic fertilisers, noting an increase in soil microbial activity that subsequently enhances crop growth and suppresses pests and diseases (Chang et al., 2010; Zhang et al., 2012). Researchers have established that tea cultivated with bio-organic fertilizers demonstrates superior colour and taste compared to tea treated with chemical fertilisers (Lin et al., 2010). Moreover, empirical studies have shown that the use of organic fertilisers results in increased seedling biomass and significantly improves the soil fungal-to-bacterial ratio and soil enzyme activity (Sun et al., 2017; Xu et al., 2010). Additionally, the application of organic fertilizers has the potential to mitigate soil acidification and consequently enhance plant yields (Li et al., 2018).

Carbon-rich biochar have numerous notable properties. These include minimal bulk density, substantial surface area, effective adsorption capacity, excellent stability, and an elevated pH (Sohi et al., 2010; Ali et al., 2019; Arif et al., 2017). Numerous studies have demonstrated that the addition of biochar to soils can alter its physicochemical characteristics. These include an increase in pH (Demisie & Zhang, 2015) and organic carbon levels (Ciampitti & Vyn, 2012), The addition of biochar has been shown to enhance soil fertility and structure (Jones & Willett, 2006), promote crop growth (Verheijen et al., 2010), reduce greenhouse gas emissions, and sequester carbon (Augustenborg et al., 2012).

Bamboo biochar, a form of engineered biochar, is generated by the pyrolysis of bamboo residue under anoxic conditions. This material has garnered considerable interest owing to its advantageous properties for absorption, as a catalyst support, and in agricultural applications (Tian, Wang & Si, 2010; Liao et al., 2013). Currently, research on the potential benefits of bamboo biochar in improving soil quality in tea gardens is lacking. This study hypothesized that bamboo biochar reduces soil acidification in tea gardens by changing soil properties, boosting enzyme activity, and altering microbial structure. Therefore, this study aimed to examine these effects when bamboo biochar is combined with an organic fertiliser, providing a theoretical basis for improving soil in tea plantations.

Materials and Methods

Land selection

The test soil was obtained from the Xin’an Village Modern Agricultural Demonstration Base, located in Dazhu Town, Suichang County, Zhejiang Province (28°13′N to 28°29′N and 118°41′E to 119°20′E). The base is located within a subtropical monsoonal climate zone. The vertical climate of the mountainous region is markedly distinct, with a mean annual temperature of 16.8 °C. The soil type is yellow-red owing to its transitional properties between yellow-brown soil and red soil (Zhang, Guo & Yan, 1984), and the predominant crop is tea, with Longjing 43 being the most prevalent variety. Soil samples were collected from a depth of 20 cm and classified to ascertain the soil background values in June 2017.

Soil treatment and collection

Five distinct treatments were implemented: a control group with no fertilizer (CK1); a control group utilizing organic fertilizer at a rate of 7,500 kg/hm2 (CK2); a treatment combining organic fertiliser at 7,500 kg/hm2 with bamboo biochar at 1,500 kg/hm2 (T1); a treatment incorporating organic fertiliser at 7,500 kg/hm2 and bamboo biochar at 2,250 kg/hm2 (T2); and a treatment that included organic fertiliser at 7,500 kg/hm2 alongside bamboo biochar at 3,000 kg/hm2 (T3). The surface area of the treatment plots was 48 m2 (6 m wide and 8 m long). Each treatment was replicated three times, resulting in a total of 15 plots. The experiment employed a randomised block design. The fertiliser was applied to the plots on 9 November 2018, 11 November 2019, and 10 November 2020. The fertilisation method entailed even fertilisation of the soil by opening trenches (trench depth of 15 cm). The bamboo biochar used in the experiment is derived from bamboo straw, which has undergone carbonisation at 800 °C and subsequent filtering through a 100-mesh sieve. Bamboo biochar exhibits the following physicochemical properties: 637.65 g/kg organic carbon, 6.32 g/kg total nitrogen, 1.35 g/kg total phosphorus, 5.13 g/kg total potassium; a pH of 9.87, a specific surface area of 257.60 m2/g, and a total pore volume of 0.21 cm3/g. Organic fertiliser and bamboo-based biochar were provided by Suichang Lujin Organic Fertilizer Co., Ltd. The samples were collected on 10 June 2021, with 20 cm soil samples taken in an S-shape in each plot.

Soil chemical index assessment

The concentration of soil organic carbon was quantified utilizing the hydrothermal potassium dichromate oxidation colorimetric method (Lu, 2020). Soil total nitrogen content was determined using the Kjeldahl distillation method (Jones & Willett, 2006). Available nitrogen was calculated using the alkali hydrolysis diffusion method (Lu, 2020). Total potassium content was determined using an FP6410 ammonium acetate extraction flame photometer (Jingke Instrument Co., Ltd., Shanghai, China) (Bao, 2000). Available potassium was quantified by NH4CH3CO2 extraction and atomic absorption spectrometry (Bao, 2000). The determination of slowly available potassium was conducted using nitric acid extraction and atomic absorption spectrophotometry (Guzel & Ortas, 1989). Total phosphorus content was determined using a three-step process involving NaOH fusion, Mo-Sb colorimetry, ultraviolet spectrophotometry (UV2600, Shimadzu, Japan) (Miranda, Espey & Wink, 2001). Available phosphorus was extracted using molybdenum anti-absorption spectrophotometry with a 0.5 mol/L NaHCO3 solution (Lu, 2020). Soil pH was determined via soil liquid extraction (water/soil = 2.5/1, v/w) using a pH meter (Bao, 2000).

Soil enzyme activities

Soil catalase activity was quantified using ultraviolet spectrophotometry with a 752 automatic ultraviolet spectrophotometer (Liu, Yi & Xia, 1996). A colorimetric method involving p-nitrophenylphosphate disodium was used to determine soil acid phosphatase activity (Dick, Cheng & Wang, 2000). Urease activity was measured by a colorimetric technique using sodium hypochlorite and phenol (Yi et al., 2021). Sucrase activity was determined using the 3,5-dinitrosalicylic acid colorimetric method (Lin, 2010). Additionally, β-glucosidase activity was determined by a colorimetric method using p-nitrophenol at 400 °C (Eivazi & Tabatabai, 1988).

DNA extraction and 16S rRNA gene amplicon sequencing

Genomic DNA was extracted from soil samples using the OMEGA Soil DNA Kit (Omega Bio-Tek, Norcross, GA, USA) according to the manufacturer’s instructions. Subsequently, PCR was conducted to amplify the V4-V5 hypervariable regions of the bacterial 16S rRNA genes, employing the universal primers 515F and 907R in a 25 μL reaction volume. The thermocycler conditions were set as follows: initial denaturation at 98 °C for 30 s; 25 cycles of denaturation at 98 °C for 15 s, annealing at 55 °C for 30 s, and extension at 72 °C for 30 s; followed by a final extension at 72 °C for 5 min. Each sample was independently amplified in triplicate, purified using Vazyme VAHTSTM DNA Clean Beads (Vazyme, Nanjing, China), and quantified using the Quant-iT PicoGreen dsDNA Assay Kit (Invitrogen, Carlsbad, CA, USA). The amplicons were pooled and sequenced using the Illumina NovaSeq platform at Shanghai Personal Biotechnology Co., Ltd.

Sequence analysis

The 16S rRNA-amplified sequences were analysed and compared with reference sequences available in the National Center for Biotechnology Information (NCBI) database. Taxonomic classifications of the samples were determined using the Basic Local Alignment Search Tool (BLAST) algorithm, and subsequent comparisons were made with entries in GenBank (Quast et al., 2012). Operational taxonomic unit (OTU) cluster analysis was performed at a 97% similarity threshold using USEARCH (version 10.0), and chimeric sequences were removed using the Denovo template within USEARCH. A uniform analysis approach was used to select the minimum number of random sequences per sample. The sequences were deposited in the NCBI Sequence Read Archive (SRA) under the accession number PRJNA1134907.

Statistical analysis

A one-way analysis of variance (ANOVA) with least significant difference (LSD) test was conducted using SPSS (version 17.0) to evaluate the physicochemical parameters, enzyme activity, relative bacterial abundance, and diversity among the samples. GraphPad Prism 8 was used to generate the figures. The significance level was set at P < 0.05.

Results

Impacts of bamboo biochar on soil physicochemical properties

Compared to the CK1 treatment, the organic fertilizer (CK2) treatment significantly increased soil pH (11.6%), available potassium (37.9%), and slowly available potassium (211.3%), and significantly decreased total potassium (6.3%), available nitrogen (23.3%), and available phosphorus (54.0%), while organic carbon, total nitrogen, and total phosphorus did not change. The combined application of organic fertiliser and bamboo biochar (T1, T2, and T3) resulted in a significant increase in pH (15.3%), total nitrogen (52.0%), available nitrogen (91.5%), total phosphorus (91%), available phosphorus (48.4%), available potassium (94.2%), and slowly available potassium (107.7%), while a notable decrease in the total potassium (49.3%) was observed when compared to the CK1 treatment. Notably, the combined treatments did not exert a considerable effect on the organic carbon content (Fig. 1).

Figure 1 The effect of bamboo biochar on the physicochemical characteristics of soil.

(A) pH; (B) organic carbon; (C) total nitrogen; (D) available nitrogen; (E) total phosphorus; (F) available phosphorus; (G) total potassium; (H) available nitrogen; (I) slowly available potassium. The data are presented as the mean ± standard error of the mean (SEM) derived from three independent samples. Statistical significance was determined using one-way ANOVA, with significant differences vs CK1 and CK2 indicated by * and #, respectively (P < 0.05).

Effect of bamboo biochar on soil enzyme activity

The CK2 treatment notably increased soil catalase (51.7%) and β-glucosidase (84.0%) activities, whereas no discernible change was observed in acid phosphatase, urease, and sucrase activities compared to the CK1 treatment. The combined application of organic fertiliser and bamboo biochar (T1, T2, and T3) resulted in a significant increase in sucrase (54.0%), catalase (68.7%), and β-glucosidase (68.4%) activities, whereas acid phosphatase activity was reduced by 52.5%. No effect on urease enzyme activity was observed (Fig. 2).

Figure 2 Bamboo biochar’s impact on soil enzyme activity.

(A–E) Urease; acid phosphatase; catalase; sucrase; β-glucosidase activities. The data are presented as the mean ± SEM derived from three independent samples. Statistical significance was determined using one-way ANOVA, with significant differences vs CK1 and CK2 indicated by * and #, respectively (P < 0.05).

Microbial community composition analysis

The most prevalent phylum in most soil samples and treatment conditions was Proteobacteria, followed by Acidobacteria, Actinobacteria, Chloroflexi, Bacteroidetes, Firmicutes, and Gemmatimonadetes. The relative abundances of these bacteria ranged from 56.69–43.00%, 25.93–12.96%, 16.69–10.10%, 6.84–3.11%, 10.13–1.41%, 8.14–0.90%, and 2.78–0.67%, respectively (Fig. 3A). The relative abundance of Proteobacteria increased significantly in the soil samples following the addition of bamboo biochar, whereas the absolute abundance of Acidobacteria decreased (Figs. 3B and 3C).

Figure 3 Microbial community composition analysis.

(A) Microbial taxonomy composition analysis; (B) relative abundance of Proteobacteria; (C) relative abundance of Acidobacteria. The data are presented as the mean ± SEM derived from three independent samples. Statistical significance was determined using one-way ANOVA, with significant differences vs CK1 and CK2 indicated by * and #, respectively (P < 0.05).

Bacterial alpha and beta diversity analyses

Alpha diversity analyses revealed a notable elevation in the Shannon, Simpson, and Pielou diversity indices in the T1 treatment compared to the CK1 treatment. However, no statistically significant differences were observed compared to the CK2 treatment. No significant differences were observed in Chao1, Faith’s, and Good’s coverage, and observed species between the CK1 and CK2 treatments (Fig. 4). Principal coordinate analysis (PCoA) and non-metric multidimensional scaling analysis (NMDS) demonstrated that the bacterial community structure of the bamboo biochar treatment group differed from that of the CK1 and CK2 treatments (Fig. 5). However, the differences between the groups were not significant according to a permutational multivariate analysis of variance (PERMANOVA) (Table 1). This may be attributed to the fact that the limited number of repetitions in the experimental design (three) may result in a reduction in the statistical efficacy of both univariate and multivariate statistics.

Figure 4 Impact of bamboo biochar on soil bacterial communities’ alpha (α) diversity.

N = 3 (*P < 0.05; **P < 0.01).

Figure 5 Impact of bamboo biochar on soil bacterial communities’ beta (β) diversity.

(A) Principal coordinates analysis (PCoA) results of β diversity; (B) non-metric multidimensional scaling analysis (NMDS) analysis of β diversity. N = 3 (*P < 0.05; **P < 0.01).

Table 1 PERMANOVA test considering beta diversity indexes.

Group1	Group2	Permutations	Pseudo-F	P value	q value	
CK1	CK2	999	4.66	0.08	0.12	
CK1	T1	999	5.10	0.09	0.12	
CK1	T2	999	4.63	0.11	0.12	
CK1	T3	999	2.93	0.12	0.12	

Sequence analysis and identification of potential microbial markers

A total of 17,820 OTUs were identified across all samples. In addition, 629 OTUs were observed in all samples. In total, 4,358 (CK1), 3,161 (CK2), 3,986 (T1), 3,037 (T2) and 3,278 (T3) OTUs were identified (Fig. 6). The addition of bamboo biochar (T1) slightly increased the overall number of OTUs. Conversely, the use of organic fertiliser reduced the total number of OTUs.

Figure 6 The number of OTUs in the bacterial community of each treatment.

A notable discrepancy was observed between the control and the bamboo biochar-treated groups, as evidenced by the linear discriminant analysis effect size (LEfSe) results. In particular, the CK1 treatment exhibited considerably higher relative abundances of Acidobacteriales, Acidimicrobiia, Frankiales, Acetobacterales, and Acetobacteraceae, whereas the CK2 treatment displayed a markedly higher relative abundance of Acidobacteria compared to the other treatments. Furthermore, the relative abundances of Deltaproteobacteria, Myxococcales and Betaproteobacteriales were substantially increased in the T1 treatment. Similarly, the relative abundances of Rhodanobacteraceae and Sphingobacteriaceae were considerably higher in the T2 treatment. Additionally, the relative abundances of Bacteroidetes, Bacteroidia, and Cytophagales were substantially higher in the T3 treatment (Fig. 7).

Figure 7 Identification of potential microbial markers.

Differential analysis of abundance of soil bacterial taxa based on LEfSe results. N = 3.

Discussion

Soil acidity is a critical determinant of crop productivity, given that >50% of the arable land and >30% of the global land area is composed of acidic soils (Meena et al., 2019). Research has indicated that tea trees acidify soil (Tongsiri et al., 2020). Biochar, a carbon-rich material, is produced through the pyrolysis of biomass in a hypoxic environment (Lehmann & Joseph, 2009). The application of biochar has been shown to elevate the pH of soils used for potted tea trees (Hailegnaw et al., 2019), corroborating our findings. A study by Wang et al. (2019a, 2019b) demonstrated that the incorporation of wheat biochar leads to elevated levels of organic carbon, total nitrogen, and available potassium. The increase in available nitrogen, phosphorus, and potassium influences soil microbial and enzymatic activities, thereby augmenting soil fertility (Alkorta et al., 2003). In the present study, the application of bamboo biochar improved the soil total nitrogen, available nitrogen, total phosphorus, available phosphorus, available potassium, and slowly available potassium contents. The application of biochar has been shown to enhance nitrogen retention in soil by reducing gaseous loss to the environment. Furthermore, it maintains phosphorous availability for plant growth by decreasing the leaching rate. However, the effects of biochar on soil composition, particularly regarding potassium and other nutrients, appear to be inconsistent and exhibit a mixed array of positive and negative effects (Hossain et al., 2020).

Soil enzymes play a pivotal role in maintaining soil health and nutrient cycling, and serve as biological indicators of soil quality (Ghani et al., 2019). The combined application of organic fertiliser and bamboo biochar resulted in a significant increase in sucrase, catalase, and β-glucosidase activities, whereas acid phosphatase activity was reduced. β-glucosidase activity initially decreased following the addition of bamboo biochar, but subsequently increased with higher concentrations of bamboo biochar. β-Glucosidase is integral to cellulose degradation, and the degradation products serve as primary energy sources for soil microorganisms (Das & Varma, 2011). In the present study, β-glucosidase activity was notably higher in the bamboo biochar treatment group. This increase may be attributed to the capacity of bamboo biochar to enhance the plant microenvironment, thereby promoting plant growth, belowground root biomass, aboveground biomass, and soil organic matter accumulation. These factors collectively provide more substrates, thereby augmenting β-glucosidase activity. Sucrase activity significantly increased following the addition of bamboo biochar, consistent with the results of previous studies (Jiang et al., 2021). The presence of sucrase in soil aids in carbohydrate conversion and it is a key indicator of soil fertility (Pajares et al., 2011). Catalase activity improved with the addition of bamboo biochar, a finding that is consistent with the results reported by Tu et al. (2020). Catalase is a pivotal soil enzyme that decomposes hydrogen peroxide in the soil, thereby preventing its accumulation and subsequent toxicity to plant roots. The incorporation of bamboo biochar enhances soil aeration and water retention, creating a more conducive environment for plant growth. Moreover, bamboo biochar can adsorb noxious substances present in the soil, reducing the inhibitory effect of these substances on soil microorganisms and enzyme activities. Soil acid phosphatase activity has been observed to decrease with the addition of biochar (Qin et al., 2020). Our findings indicate that acid phosphatase activity was significantly reduced following the application of bamboo biochar, which may be attributed to the increase in soil pH. Acid phosphatase activity is closely related to substrate availability.

The physicochemical properties of soil influenced by biochar can affect microbial biomass and community structure (Smith, Collins & Bailey, 2010). Variations in biochar dosage and soil type are likely to be the primary factors affecting the composition and structure of soil microbial communities (Khodadad et al., 2010). The incorporation of biochar into nutrient-deficient soils can enhance biomass and improve the nutrient utilization efficiency (Alburquerque et al., 2014). Biochar incorporation enhances the overall diversity of soil bacterial communities (Zhang et al., 2017). Specifically, Kolton et al. (2017) reported that biochar application can increase bacterial diversity indices, such as the Shannon index. Additionally, Ali et al. (2019) found that the inclusion of biochar increased Shannon and Simpson indices. Few studies have examined the effects of bamboo biochar on the soil bacterial community structure. Our study demonstrated that the Shannon, Simpson, and Pielou diversity indices exhibited significant increases following bamboo biochar treatment. Furthermore, the abundance of Proteobacteria markedly increased with the addition of bamboo biochar, whereas that of Acidobacteria significantly decreased. Research indicates that Proteobacteria readily dominate environments with abundant active organic carbon (Lewin et al., 2016; Sun et al., 2016). The application of bamboo biochar, which enhances the availability of soil active organic carbon, led to an increased relative abundance of Proteobacteria, Actinobacteria, and Firmicutes. Conversely, Acidobacteria, which thrive in anaerobic conditions, may experience a decline in abundance due to the enhanced soil aeration facilitated by bamboo biochar (Yamada & Sekiguchi, 2009). Taken together, these alterations caused significant changes in the bacterial community. This elucidated the impact of bamboo biochar on soil microbial diversity in tea gardens to enhance our understanding of its ameliorative effects on acidified soils within these ecosystems.

Conclusions

Soil acidification in tea gardens represents a critical challenge that can hinder soil succession processes. Our findings demonstrated the effectiveness of bamboo biochar in alleviating soil acidification in tea gardens. This study examined the influence of bamboo biochar on the physicochemical properties, enzymatic activity, and microbial diversity of tea garden soil, providing a substantial theoretical basis for the large-scale application of bamboo biochar as a soil acidification remediation strategy.

Supplemental Information

Supplemental Information 1 Soil chemical index test raw data.

Supplemental Information 2 Raw data of soil enzyme activity detection.

Additional Information and Declarations

Competing Interests

Author Contributions

DNA Deposition

Data Availability

The authors declare that they have no competing interests.

Si-Hai Zhang performed the experiments, prepared figures and/or tables, authored or reviewed drafts of the article, and approved the final draft.

Yi Shen performed the experiments, analyzed the data, prepared figures and/or tables, and approved the final draft.

Le-Feng Lin performed the experiments, analyzed the data, prepared figures and/or tables, and approved the final draft.

Su-Lei Tang performed the experiments, analyzed the data, prepared figures and/or tables, and approved the final draft.

Chun-Xiao Liu analyzed the data, prepared figures and/or tables, and approved the final draft.

Xiang-Hua Fang analyzed the data, prepared figures and/or tables, and approved the final draft.

Zhi-Ping Guo conceived and designed the experiments, authored or reviewed drafts of the article, and approved the final draft.

Ying-Ying Wang conceived and designed the experiments, authored or reviewed drafts of the article, and approved the final draft.

Yang-Chun Zhu analyzed the data, prepared figures and/or tables, and approved the final draft.

The following information was supplied regarding the deposition of DNA sequences:

The microbiome 16S rRNA libraries are available at NCBI SRA: PRJNA1134907.

The following information was supplied regarding data availability:

The raw measurements are available in the Supplemental Files.

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
