# Peer review of "Effects of bamboo biochar on soil physicochemical properties and microbial diversity in tea gardens"

_PeerJ, doi:10.7717/peerj.18642_

## Round 0.1 · original submission · Major Revisions

Dear authors,

One reviewer (R2) suggested Rejection while the second said "Major Revisions". Please read their comments and suggestions carefully.

Being unable to find sequence depositions is serious. Please make sure you provide a link to them in your Methods.

Pay attention to the call for PERMANOVA.

Submitting papers without enough proofreading annoys reviewers and can impact authors' reputations within the field. If you resubmit, please go over the English wording very carefully and make sure there are no formatting errors in references, no missing explanations of symbols like "*" and "#".

One of the most important criticisms is "Lines 81-82: You only had 3 replicates contrary to international recommendations for microbiome studies." With 3 replicates, there are only 2 degrees of freedom, making statistical testing very difficult. If you can do extra replicates, please do so. If not, please explain in your rebuttal why not and what steps you have taken in the statistical analysis and the wording to compensate for this deficiency.

Reviewer 1 ·

Basic reporting

Significant work would have to be done to make this paper publishable. The authors need to clean up the manuscript and resubmit it.
The paper deals with important topics but has not been proofread adequately
Overall, the grammatical errors and unfinished sentences take away from the manuscript (authors need to do their due diligence).
-Some sentences are overly long and redundant. I suggest that the authors rewrite these sentences for better readability and conciseness.
-Removing unnecessary words and improving sentence structure will help streamline the text.
-Remove the extra punctuation like you used commas several times in a single sentence.
The study hypothesis is missing
-Several references and citations are out of format.
Abstract
-Line 24-27: how much increased as compared to control? give a comparative percentage of increase or decrease.
- How much increase or decrease did you find in the result, provide the digit or percentage, it will show your study's importance otherwise it will be unclear.
Introduction
Line 42-44: Provide reference
Line 46: Provide reference
Line 49-50: Rephrase the sentence for a better understanding

Experimental design

The experimental design needs improvement. The author needs to rewrite it. I recommend the author to look at some good related papers get ideas from them and rewrite strong methodology. It is important for the reader to of interested if they have the same area of research or if someone wants to perform a similar analysis.
Line 72: According to which soil classification?
Line 73-73: two depths? 0 cm and 20 cm? clarify it
Line 87: provide the charcoal chemical and physical properties.
Line 91-92: what are you saying here, I don't understand. Rewrite this sentence for better understanding.
Line 94: Provide References
Line 07-100: Please provide the reference of each method

Validity of the findings

Results
Results need to improvement and maintain clarity. Removing unnecessary words and improving sentence structure will help streamline the text.
-In results, first, always give the full form of acronyms and then use these acronyms in results and discussion.
-Result look like imaginary. if your study has significant results then provide them in digits or percentages and compare them with each other.
-How much increase or decrease enzyme activities?
Discussion
-Discussion needs strong diligence to rewrite.
-what is the significance of your results, explain them..I don't see any reflection of your study's importance after your results are discussed. Rewrite your discussion logically to support your results. What gaps are your research results filling?
Conclusion
It is very general, you already discuss this in your abstract and results. Conclusions aren't simply an overview of a paper. Instead, they should reiterate why your research is important. How it can fill the existing research gap in that area? If done well, conclusions can leave readers feeling both satisfied and hungry for more.

Additional comments

There are significant issues with the English language. It would be beneficial to have a native English speaker review the manuscript for clarity.

Annotated reviews are not available for download in order to protect the identity of reviewers who chose to remain anonymous.

Reviewer 2 ·

Basic reporting

The English language should be improved to ensure that an international audience can clearly understand your text. Some examples where the language could be improved
include lines 27-28, 31, 34-36, 51, 61, 70, 73-75, 91-94, 117, 138-190121, 128 – the current phrasing makes comprehension difficult. I suggest you have a colleague who is proficient in English and familiar with the subject matter review your manuscript, or contact a professional editing service.

Experimental design

While the study in the manuscript by Zhang et al. is within the aims and scope of the journal, it is seriously flawed in several places.
1. Methods were not properly described in order to be replicated
2. Incorrect description in several places showing the authors might lack expertise on the subject matter. See for instance lines 115-116 where MiSeq reagent kit v3 was incorrectly described as used for DNA extraction
3. The low number of replicates (3) is unacceptable. See Nannipieri et al 2019 about Recommendations for soil microbiome analyses.

Validity of the findings

The manuscript in it's current version is not replicable as several details are lacking.
There was mention of data deposition in NCBI but no link on how the data can be accessed.
The conclusions are bogus and I find it hard to link it with the results.

Additional comments

The title needs revision for clarity.
Line 64: Here you just mentioned organic fertilizer without introducing it earlier.
Lines 81-82: You only had 3 replicates contrary to international recommendations for microbiome studies. Also, use of terms like 'base' and 'cell' are not easily recognisable by international audience.
Lines 107-113: Not described in details for reproducibility
lines 122-127: See for instance Gloor et al., 2017 for proper protocol on how to process amplicon sequence data
Lines 126-127: Sequence deposition was mentioned but no more details on how to access it.
Line 166: should have been 'bacterial alpha and beta diversity analyses
Lines 171-175: This was not accompanied with multivariate statistics like PERMANOVA anywhere in the manuscript
Lines 176-190: It is very confusing as you mentioned OTUs in the methods and here you are writing about ASVs.
Line 197: You suddenly mentioned compost here which was never introduced.
The descriptions of figures 1 and 2 is ambiguous especially the * and #. What do the error bars represent?
Figure 3 should have been bars for each treatment group and not for each sample unit.

---

## Round 0.2 · Minor Revisions

Dear authors,

Only some small changes are needed now. Please make changes suggested by reviewer 2 and the article will be ready for publication.

Reviewer 1 ·

Basic reporting

The authors have incorporated all the basics suggested in the revised manuscript.

Experimental design

Satisfied

Validity of the findings

Satisfied

Reviewer 2 ·

Basic reporting

The authors improved the paper with regards to the language editing. However, terms like biochar should not have been replaced with charcoal

Experimental design

The experimental design is still flawed with regards to the number of replication (n=3) for microbiome studies. That was why even though they were 'visual' differences between the groups, PERMANOVA failed to detect the differences as the authors presented in Table 1. I suggest the authors note this limitation in the manuscript with regards to the interpretation of the microbiome data. The sample size was insufficient to draw any meaningful conclusion

Validity of the findings

All data were provided

Additional comments

Change charcoal to biochar. Make the limitation of the study clear to the readers.

---

## Round 0.3 · Minor Revisions

Thanks for addressing reviewer concerns. The article is close to ready for publication. A PeerJ Section Editor has asked for some changes: "The authors should explain how the biochar is produced. The initial statement "Biomass leads to biochar" is not accurate. In addition, the pyrolysis of bamboo that leads to biochar, and some quality criteria (such as particle size) should be included."

---

## Round 0.4 · accepted · Accept

Thanks you for making the requested changes. The text reads more clearly now.